# Clustering-Based Decision Tree for Vehicle Routing Spatio-Temporal Selection

**Yixiao Liu** [1], **Lei Zhang** [2,*], **Yixuan Zhou** [2], **Qin Xu** [1], **Wen Fu** [1] **and Tao Shen** [3]

[1] Faculty of Data Science, City University of Macau, Macau 999078, China; d21091100431@cityu.mo (Y.L.); d19091105370@cityu.mo (Q.X.); d19092105124@cityu.mo (W.F.)

[2] Department of Traffic Information and Control Engineering, Tongji University, Shanghai 200092, China; jokerzyx@tongji.edu.cn

[3] Faculty of Information Engineering and Automation, Kunming University of Science and Technology, Kunming 650506, China; shentao@kust.edu.cn

[*] Correspondence: reizhg@tongji.edu.cn

**Abstract:** The algorithm of the clustering-based decision tree, which is a methodology of multimodal fusion, has made many achievements in many fields. However, it is not common in the field of transportation, especially in the application of automobile navigation. Meanwhile, the concept of Spatio-temporal data is now widely used. Therefore, we proposed a vehicle routing Spatio-temporal selection system based on a clustering-based decision tree. By screening and clustering Spatio-temporal data, which is a collection of individual point data based on historical driving data, we can identify the routes and many other features. Through the decision tree modeling of the state information of Spatio-temporal data, which includes the features of the historical data and route selection, we can obtain an optimal result, that is, the route selection made by the system. Moreover, all the above calculations and operations are done on the edge, which is different from the vast majority of current cloud computing vehicle navigation. We have also experimented with our system using real vehicle data. The experiments show that it can output path decision results for a given situation, which takes little time and is the same as the approximated case of networked navigation. The experiments yielded satisfactory results. Our system could save a lot of cloud computing power, which might change the current navigation systems.

**Keywords:** clustering-based decision tree; Spatio-temporal data; navigation system

## 1. Introduction

As a result of industrialization and migration from large cities and increasing population, traffic in urban areas has risen [1]. This increase in the number of vehicles causes delays for drivers' trips, and, therefore, navigation has become a very important driving concern [2]. Nowadays, almost every household's vehicle is equipped with an onboard or a portable GPS navigation system. The system works by locating its vehicle and receiving and sending data to the cloud. The current GPS positioning algorithm is based on Spatio-temporal data for real-time operation on a large scale [3]. The cloud calculates the real-time road profile by sending the data information of the point through countless terminals and provides a real-time latest route to the terminal by issuing navigation instructions to avoid congested road sections [4]. From this, we can see that the pressure on the cloud to receive, send, and process real-time data is very great, especially during peak traffic periods [5].

With the increasing number of vehicles, the timeliness (delay) and bandwidth limitations of device processing naturally limit the Spatio-temporal-based cloud computing model [6]. Thus, edge computing, which is on a smaller scale, has now become a new hot topic [7]. By extending data and computing to the edge, network delay no longer poses a problem and optimal decisions can be made promptly [8].

At the same time, the route choice is often not the shortest route between the starting point and the ending point. It might be a route that takes a longer distance but a shorter journey time. Different route decisions are also made for different periods or different weather conditions for the same starting and ending point. Thus, a good route decision model can be obtained by learning from these past route decisions with drivers' experience. Data on the edge side obtain the Spatio-temporal features that the model needs.

Therefore, we proposed a clustering-based decision tree for vehicle routing Spatio-temporal selection. Compared with existing offline route decisions, our system better reflects the Spatio-temporal complexity of the road network and does not impose excessive restrictions on the prerequisites for route decisions. By screening and clustering Spatio-temporal data, which is a collection of individual point data based on historical driving data, we can identify the routes and many other features. Through the decision tree modeling of the state information of Spatio-temporal data, which includes the features of the historical data and route selection, we can obtain an optimal result. Through edge computing, we can directly choose a route by processing historical data at the edge, thus greatly reducing the pressure of data transmission and processing between the cloud and the terminal during navigation. In this way, the cloud can be liberated to handle other more important work.

In the following sections, related works, the principle of our system, the application of our system to real-world datasets, and the presentation and comparison of the results will be presented, respectively.

## 2. Related Work

### 2.1. Clustering-Based Decision Trees

Different researchers used various descriptive methods for combining clustering and decision tree algorithms [9]. However, the common aspect is that these methods are usually called "classification by clustering," and the algorithms used are very specific (such as K-means + ID3). In general, evident benefits from grouping in the classification have been established [10]. In different domains, clustering is used to distinguish the features of different datasets and decision trees are used in different outcome-oriented modeling.

For example, in the field of network and information security, researchers have proposed a method of supervised anomaly detection [11]. This "K-means + C4.5" method was developed by cascading two machine learning algorithms (K-means clustering and a C4.5 decision tree). In the first stage, the region of similar instances represented by the K-means cluster is solved based on the Euclidean distance between the computed data and their cluster centroid. Then, the K-means are cascaded to the C4.5 decision tree by using the instances in each K-means cluster to model the decision tree. From another research study, it was learned that cascading machine learning algorithms can provide better results than machine learning alone [12]. In addition, other researchers have used the "K-means + ID3" method to cascade K-means clustering and ID3 decision tree learning methods to identify and classify abnormal and normal activities in computer networks [13], mechanical systems [14], and electronic circuits.

In addition, in the field of education, researchers have also used such methods in student databases [15]. They used a K-means clustering algorithm and a decision tree algorithm for data mining to predict students' learning activities. This work is advantageous for both teachers and students. With the help of highly accurate prediction results, it can help teachers take appropriate measures at the right time to improve the quality of their teaching. This can improve students' performance and reduce failure rates.

In the field of health, researchers have also combined the K-means clustering algorithm with the decision tree algorithm [16]. Through these research results, we found that the accuracy of aggregated K-means clustering, and decision tree algorithms are better than that of other algorithms, such as the genetic algorithm, classifier training algorithm, and classification algorithm based on a neural network, in the proposed application to spirometry data [17]. Other researchers have also proposed a new hybrid medical expert system for effective medical diagnosis [18]. Their proposed hybrid system consists of two

efficient algorithms: the genetic-based decision tree algorithm and the weighted K-means clustering algorithm.

### 2.2. Spatio-Temporal Platform and Navigation Route

Spatio-temporal data, to a certain extent, is the user location and trajectory in-formation data collected through the Location-based service [19]. Because of its Spatio-temporal precision and large-scale user coverage, it has high commercial value and scientific research value [20]. To achieve rapid response and massive data support capability of the cloud, edge computing is gradually used in the Spatio-temporal platform in various fields [21–23]. With the push from cloud services, the pull from the Internet of Things (IoT), and the change from data consumers to data producers, we increasingly need edge computing [24,25]. Edge computing refers to a new computing model in which computing is performed at the edge of the network [26–28]. It offers several advantages. First, the edge computing model migrates part of the computing tasks in the original cloud computing center to the vicinity of the edge data source [29,30]. Moreover, compared with the traditional cloud computing model, the edge computing model has additional advantages in terms of the characteristics of big data, namely, volume, velocity, and variety [31]. Therefore, it is necessary to better design edge devices and key supporting technologies for data security based on edge devices to meet the reliability, security, and privacy protection services in the edge computing model [32].

Most of the existing route planning research is still based on cloud real-time data processing [33]. Some researchers used algorithms to simulate and predict routes. Some of them are based on pedestrian trajectory prediction learning [34], some are based on super-vised learning of experienced drivers' route choices [35], and some are based on modeling and learning of relevant information of cyber-physical systems at given points [36]. In addition, researchers have done a lot of work on offline route decision methods. Some researchers model the road network information as a circuit board with many resistors and find the channel with the lowest resistance for route selection by local current comparison methods [37]. However, a road with a large changing traffic flow cannot simply be viewed as a constant value of resistance; it changes due to a range of other factors such as time of day and weather. Some researchers have also investigated how to route buses or feeders with a given origin and destination as well as scheduling arrangements [38]. This method is mainly used for company shuttles or shuttle buses in cities and is not applicable to car navigation that may have any starting or ending point. Through research, we found that navigation routes, even under the influence of real-time traffic data, have a strong similarity to historical data [39]. Moreover, the period and weather conditions will have a great influence on navigation data of given starting and ending points [40]. Therefore, we should make full use of the Spatio-temporal data platform to build a navigation computing system to solve navigation problems more efficiently.

## 3. Methods

### 3.1. Spatio-Temporal Data Platform

The connection and modeling of the Spatio-temporal data platform are shown in the following figures. What we use in our daily travel is an on-board or a portable GPS navigation. As terminal devices, mobile phones or vehicle GPS cannot store a large amount of data, which is often stored in the cloud, nor do they have strong computing power. Therefore, the number of visits to the cloud is huge.

To reduce the cloud computing pressure, the urban traffic information platform, which is based on IoT, gradually uses edge computing to achieve rapid response and massive data support capability of the cloud. Therefore, as shown in Figure 1, we add a layer of edges. A single edge layer covers a certain scale of land. We connect the cloud–edge-terminal layer so that the computational pressure is unloaded to the edges, and then the Spatio-temporal data platform is modeled.

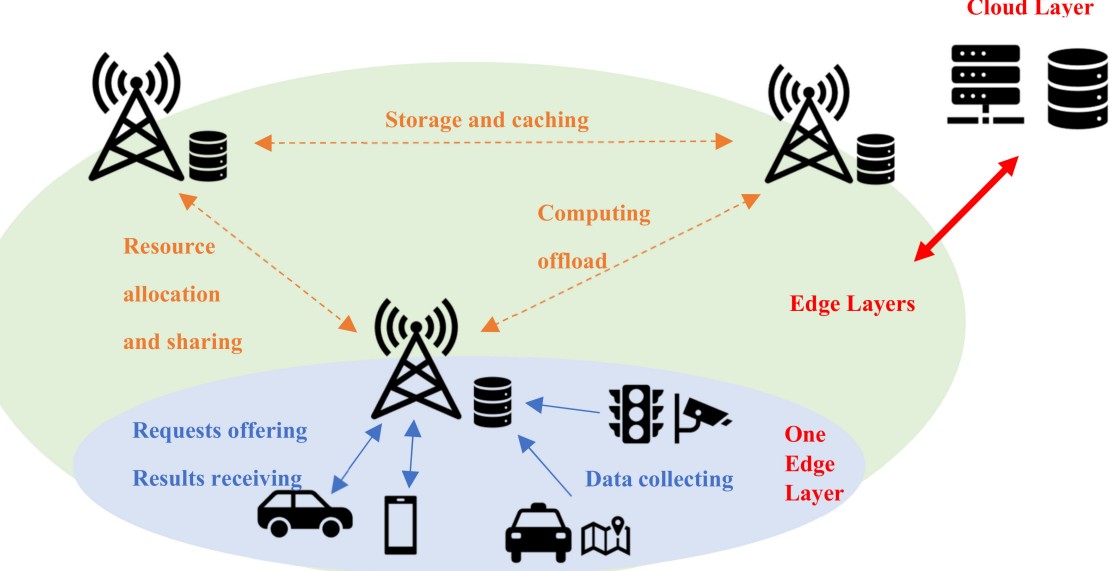

**Figure 1.** Spatio-temporal data platform.

When the whole city is divided into adjacent edge layers, the Spatio-temporal data platform can play its role well. Each edge layer can collect the Spatio-temporal data information of the vehicles in the area it is responsible for. The edge uploads this information to the cloud and stores it locally. In this way, data are read through the interaction between the edge layers. As shown in Figure 2, assuming that the two dark black edges in the upper left corner and the lower right corner are the edge ranges of the departure point and the arrival point respectively, it is possible to read in which period a vehicle has traveled from the departure point to the arrival point through the interaction between these two edge layers. Then, according to the read period and vehicle information, the Spatio-temporal data of those vehicles are interactively read to the surrounding adjacent edges until the historical Spatio-temporal data stored on the edges of all paths are obtained. In this way, only all the historical vehicle path information of the given starting point and the arrival point is read through the interaction between the edge layers. Through the analysis of this Spatio-temporal data, better path decisions can be made.

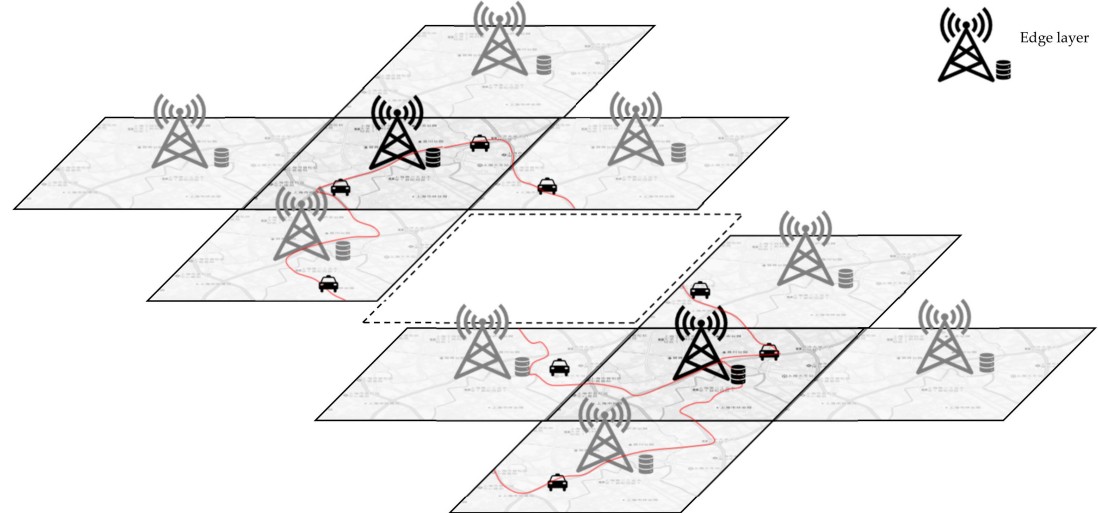

**Figure 2.** Spatio-temporal data reading.

### 3.2. Clustering-Based Decision Tree Route Selection

In this section, we present the proposed method to obtain historical drivers' experience, which leads to better results for clustering-based decision tree route selection. The method consists of four stages: data preparation, clustering, decision tree design, and interpretation of results, as depicted in Figure 3.

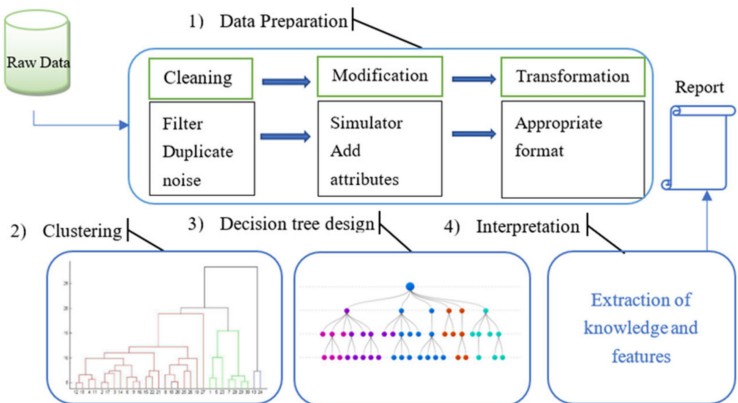

**Figure 3.** Modeling process.

### 3.2.1. Data Preparation

The first step is data preparation. This includes cleaning the dataset through the process of removing duplicate records and eliminating noise data and national travel data, extracting the required starting and ending locations and data information under the influence of personal preferences, and converting data into a format specific to data mining processing techniques.

### 3.2.2. Clustering

The main function of clustering is to cluster the routes of each piece of data after reading the required data to determine how many output results there are.

The specific implementation method is the following: Mark the coordinates of the given starting point and the target point and divide the grid on the map as the two endpoints. Number each grid and represent all the collected valid data on the grid to get a route diagram, as shown in Figure 4.

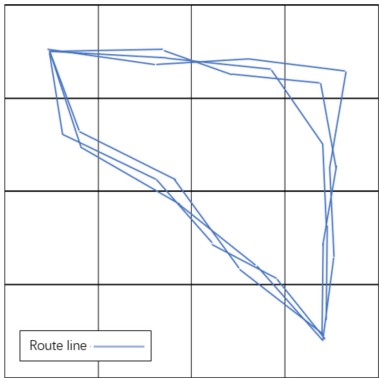

**Figure 4.** Dot connection diagram (raw).

Count the number of the grids passed by each route into a new variable. This gives us a new feature of this piece of data. Clustering this feature of all valid data can identify how many routes there are, as shown in Figure 5. The maximum number of categories is not set for clustering results. However, if the proportion of a certain type of result is less than 1%, it will be discarded to improve the accuracy of clustering results.

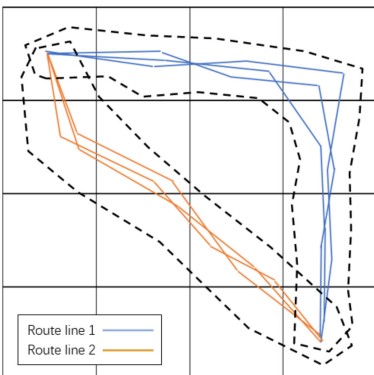

**Figure 5.** Dot connection diagram (clustered).

There are several common clustering methods as follows: divisional clustering, hierarchical clustering, density clustering, model clustering, and spectral clustering. Since we are using a large Spatio-temporal traffic dataset, which consists of many individual point data containing temporal information, spatial information, and other valid information, we chose to use the most appropriate divisional clustering for this dataset. The K-means algorithm is representative of the divisional clustering algorithm. It has the advantages of being simple and efficient even for large data sets, low time complexity, low space complexity, etc. It can perfectly accomplish the ideas we need to achieve.

After determining the starting point and destination, we will have several planned routes, which are zigzag lines. However, the historical data of automobiles are coordinates of time and geographical location, which are a collection of points. Because of the difference in receiving time and receiving frequency, the information of coordinate points recorded by cars will not be the same even if they travel on the same route. Therefore, clustering the data as described above can solve the problem well.

By observing the clustering results, we can distinguish the characteristic information of vehicles on the grid map. Since each grid represents a geographical area, the feature that a car has a grid represents that the car has passed through that area, so the clustering results can distinguish the vehicles that travel through the same area. Further, all the categories of the clustering results are all the different routes at the starting and ending points.

### 3.2.3. Decision Tree

The third step is the decision tree design stage, the goal of which is to design a set of decision trees to solve the route selection of the corresponding starting and ending points by filtering the refined attributes of the stage. In this case, we chose C4.5, which is a widely accepted algorithm in the scientific community.

This step is very important to make visible the driving experience hidden in the clustering results of the dataset. By analyzing the coordinates of the vehicles in the dataset, we can obtain additional data features that can demonstrate driving experience. For example, it is possible to analyze whether the coordinates pass through elevated roads to distinguish between different preferred route decisions. In addition to the spatial information representing vehicle coordinates in the dataset, there is also temporal information representing real-time time. Our analysis of this temporal information can also reveal some additional data features. For example, drivers tend to avoid some congested roads in the morning when making route decisions but make other decisions at other times when traffic is not heavy, such as midday.

Before the decision tree, the user can select subjectively controllable options such as whether to use the overhead. According to the user's selection, the backend filters out the available data that meet the condition in the valid data. Then, the decision tree calculation is performed according to objective conditions (such as whether the period is a peak period, whether it is a weekend, the weather conditions, etc.). By taking the state information of objective conditions as the input and the route as the output, the decision tree can be

modeled and generated. According to the decision tree model and the current state of the user, the decision result is generated, and the route can then be selected.

To solve the problem of hyperparameters, some parameters need to be pre-set or adjusted in the decision tree modeling process. Due to the huge amount of sample data, the feature classification criterion "splitter" is chosen as "random". The maximum depth of the decision tree "max_depth" is adjusted to the number of input features. The minimum number of samples required for internal node subdivision "min_samples_split" is set to 10 to avoid the influence of some maverick drivers' route choices on the results. The maximum number of leaf nodes "max_leaf_nodes" is set to the square of the number of input features to prevent overfitting.

### 3.2.4. Interpretation of Results

The route selected by the result generated by the decision tree model is not a route that fits the map. Therefore, we need to decompile the routes. We can read the entire period data of the historical vehicle formation data of the first car driving through the route in the model. By marking its entire positioning point on the map, we can then display the complete route we need.

At this point, the whole process is over. Its pseudo code is shown in the Algorithm 1 below.

---

**Algorithm 1** Prediction Model

---

**Input:**

　　Given historical data: $X[vn, loc, t, st] = \{X_1, X_2, \ldots, X_n\}$. $vn$ is vehicle number; $loc$ is geographical location, $t$ is time, and $st$ is the status of passengers.
　　Given a gridded map with every grid numbered: $M[g1, g2, \ldots, gm]$.
　　Given a prediction task: $R[sp, ep, t]$. $sp$ is the start position; $ep$ is the ending position.

**Output:**

　　The prediction model and the results of the prediction.

1: for $i \in vn$ do
2:　　$X_{vn}[loc, t, st] = X[vn, loc, t, st]$ //Divide the data according to vehicle number.
3: end for
4: for $i \in vn$ *and* $j \in t$ do //Extract data with the given start and ending position
5:　　if $X_i[loc, j][st] <> X_i[loc, j+1][st]$ *and* $X_i[loc] = R[sp]$ then
6:　　　　for $k \in (j, t)$ do
7:　　　　　　if $X_i[loc, k][st] <> X_i[loc, k+1][st]$ then
8:　　　　　　　　if $X_i[loc] = R[ep]$ then
9:　　　　　　　　　　$Data = Data + \{X_i[loc, j, st], X_i[loc, j+1, st], \ldots, X_i[loc, k, st]\}$
10:　　　　　　　　end if
11:　　　　　　end if
12:　　　　end for
13:　　end if
14: end for
15: for $i$ in $Data$ and $j$ in $M$ do //Record the grids through which each data passes
16:　　$Data_i\left[route_{feature}\right] = M_j$
17: end for
18: while //Cluster the new feature of all data to calculate the route selections
19:　　Cluster $Data_i\left[route_{feature}\right]$
20:　　$Data_i[route] = result\ of\ the\ cluster$
21: end while
22: while //Train the model
23:　　Train the $Data_i$ with the Decision Tree model
24: end while
25: route = $Data_i[R]$ //Input the task into the model to get the result
26: **Return:** The model result and the predicted results for the given task.

---

3.2.5. Limitations

Our system will also have some limitations. First, the system cannot make decisions for paths that are too short. If the given start and ending points are all within the coverage of one edge layer, the distance is too short for the system to distinguish the paths. Second, the system is unable to react to unexpected road condition information. Since the system makes decisions based on historical data analysis, some unexpected traffic conditions, such as a road is no longer allowed, will probably affect the accuracy of the system's judgment.

**4. Experiments and Results**

We used the data of all Shanghai taxis from the Didi Company for a total of two weeks from 7 to 21 April 2018. The dataset includes vehicle number, empty vehicle status, reception time, latitude and longitude coordinates, velocity, direction, braking status, and overhead status. The data density is 10 data transfers/minute/vehicle. The size of each piece of data is about 100 Bytes, and the size of our total data set reaches 120 GB. The dataset only shows Spatio-temporal information and status information of each taxi. No human driver feedback is included. All the work below is based on this dataset. Due to non-disclosure agreements, we only provide overview information of the dataset as described above. We would like to thank Professor Lei Zhang, DiDi Global Inc. (Beijing, China), and the National Transportation Information Co., Ltd. (Beijing, China). for providing the data to us in order to complete this research.

Now, consider the following example: It is the weekend, the weather is good, and we want to go to Xinzha Road from Shanghai West Railway Station but do not want to take the elevated road. Let us see what route we will get through the algorithm. The program will be implemented in the Python language.

*4.1. Utilization of Edge Computing*

We store the dataset in an edge computer and send commands to it with a mobile phone. Edge computers can download the latest data from the cloud at a specific time every week to ensure the timeliness of the data. The dataset and its operation can be realized in the edge computer. Therefore, all the data processing can be realized by edge computing, which will involve too much interaction with the cloud, thus reducing the pressure on the cloud.

In this experiment, we store the database on a computer, labeled PC1, and treat it as if it were in the cloud. Let us use another computer, labeled PC2, as the edge in this experiment; PC2 can download data from PC1. At the same time, we use a mobile phone as a terminal. We send instructions to the edge through the mobile phone. After that, the edge directly calculates and processes the data through the instructions sent by the terminal and the data downloaded from the cloud, and then it directly sends the results back to the terminal mobile phone. In this process, only the side end downloads data from the cloud, and there will be no access to the upload cloud or the process that requires cloud computing. In response to the experiment we designed, except for downloading data from PC1, PC2 did everything by itself.

*4.2. Data Filtering*

Due to the huge amount of data, we need to trim the dataset before we can process it. We first clean and delete the missing data. Then, we take the first datum every three minutes according to the time in the dataset, and we only take the first datum of each vehicle in that minute, so that we can lighten the whole dataset by a factor of nearly 200.

By reading the empty car status in the data, we know when and where the car is carrying passengers or letting passengers off by comparing before and after data. Therefore, we can filter out the data of all the starting and ending points in all the datasets for the given starting and ending points. This provides us with numerous time series of vehicle travel.

Through the date of each piece of data, we know whether the datum is on the weekend, which can be recorded as the first feature point. Through the time of each piece of data, we can know whether the datum is in the peak period, which can be recorded as the second feature point. Each piece of data has the feature information of whether it is on the elevated road, and we record whether it has been on the elevated road in each piece of data as the third feature point. By searching historical weather data, we can get the weather information at that time. We record the weather information of each piece of data as the fourth feature point. So far, we have four pieces of feature information. These four pieces of feature information will also be used as the data input to the subsequent decision tree algorithm.

### 4.3. Implementation of Clustering

4.3.1. Division of the Grid

Let us construct a grid for the map first. If the grid is too coarse, the final number of routes will be too concentrated. As a result, it will be difficult to distinguish the differences among different routes, which leads to poor clustering results. If the grid is too fine, the routes will be too scattered. However, when the number of routes is clustered, this will lead to too many routes. Moreover, sometimes the same route will be judged as two routes in the computation, owing to the difference in reading time and location, which greatly affects the accuracy of the algorithm. Therefore, the grid division needs to be like the time interval of data reading. Here, we take 0.01 longitude and 0.01 latitude as the side length of the grid to divide Shanghai into grids and number the grids, accordingly.

4.3.2. Clustering

Let us cluster the data we need. First, we mark all the collected data on the gridded map, as shown in Figure 6. (For the sake of readability, in Figure 6; Figure 7, we show the data of both the starting and ending points on the map from 6:00 a.m. to 12:00 a.m. on 7 April 2018).

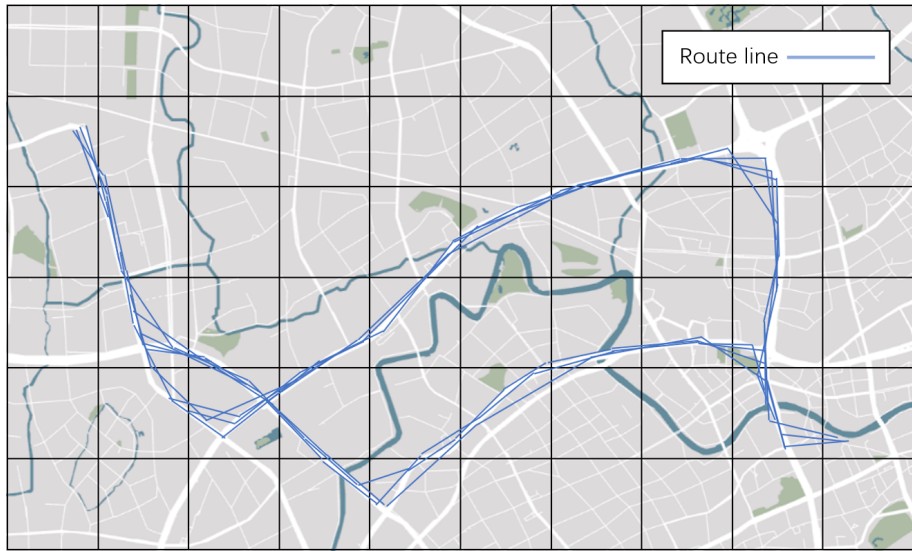

**Figure 6.** Historical roadmap (raw).

Next, we cluster the data by using the K-means clustering algorithm. After clustering, two routes are obtained, and the results are shown in Figure 7.

The route (route 1 or 2) corresponding to each piece of data is recorded and included in the characteristics of the piece of data. This data feature is the output data feature in the subsequent decision tree algorithm.

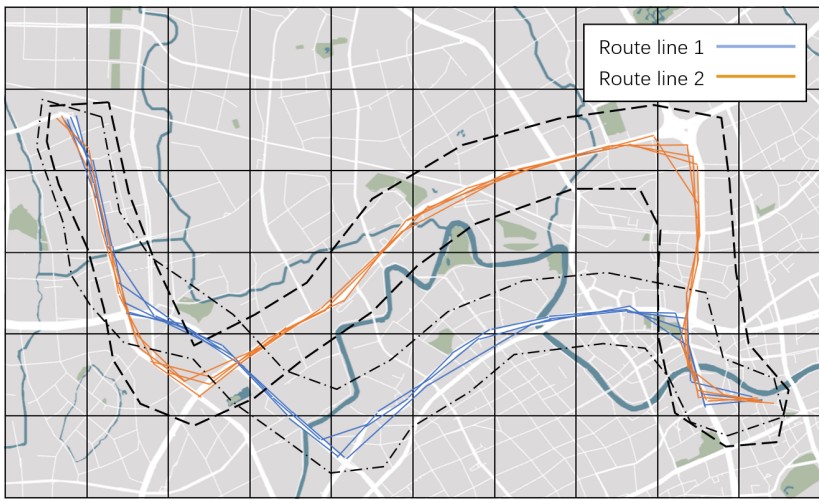

**Figure 7.** Historical roadmap (clustered).

### 4.4. Decision Tree Generation

For the processed data, the input indicates whether the collection time is during the rush hour, whether it is on the weekend, whether the vehicle passes through the elevated road, and whether the weather is sunny, and the output is the route. Figure 8 shows the result of the decision tree generation.

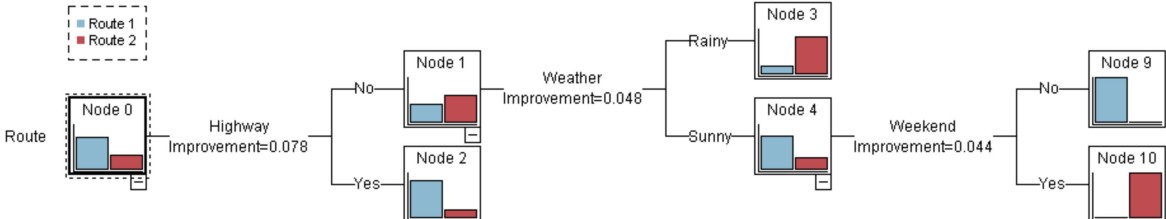

**Figure 8.** Decision tree result.

The C4.5 decision tree algorithm is used for this model. The C4.5 algorithm mainly calculates and compares the information gain ratio of each feature of the data to determine which feature has a greater impact on the results. In this model, if one of the results of the information gain ratio segmentation of a certain feature in one step has a result that the proportion of data selecting a certain route is greater than 90%, then the calculation of this branch is ended, that is, it is pruned. For example, in the result of our task, the feature with the highest decision information gain ratio for the first time is the choice of the highway. After deciding on the highway, we find that more than 90% of the historical data using the highway have traveled on route 2. We directly prune this branch and do not continue to calculate, and route 2 is the result of this decision route. Through this rule, we calculate all the data to get the decision tree result.

### 4.5. Comparison and Interpretation of the Results

In addition to the decision tree algorithm, we also performed the modeling calculation on the dataset using a support vector machine (SVM) and logistic regression. Further, 90% of the dataset was used as the training set, and the remaining 10% was used as the test set to test and compare the models. The dataset was shuffled before the training and each model was trained 10 times to increase the persuasiveness of the experiment. To evaluate the performance of each algorithm, four standard metrics were used to measure the difference between the actual route $r_i$ (always set as 1) and the prediction route $\hat{r}_i$ (set as 0 if is wrong and as 1 if is correct), including Root Mean Square Error (*RMSE*), Receiver Operating Characteristic (ROC) and Area under ROC (*AUR*), F1-score (F1), and explained Variance (*VAR*),

defined as follows, $RMSE = \sqrt{\frac{1}{n}\sum_{i=1}^{n}(r_i - \hat{r}_i)^2}$, $AUR = \frac{\sum_{ins_i \in positiveclass} rankins_i - \frac{M\times(M+1)}{2}}{M\times N}$, $F1 = 2 \times \frac{precision\times recall}{precision+recall}$, and $VAR = 1 - \frac{var(R-\hat{R})}{var(R)}$. *RMSE* measures the prediction error, and the smaller the better. AUR intuitively reflects the classification ability expressed by the ROC curve, and its value closer to 1 represents its better classification ability. The value of F1 is the summed average of precision and recall, and the larger the number of F1 values from 0 to 1, the better the achievement. *VAR* calculates the correlation coefficient, which measures the ability of the prediction result to represent the actual value [41], where $var(\cdot)$ is the variance function. Note that for *VAR*, the larger the better.

All the training was done with scikit-learn in Python. Decision tree used C4.5, which was introduced earlier. In the parameter setting of SVM, the penalty parameter C of the error term was set as 1, the kernel was set as Radial Based Function (*RBF*), and gamma was set 1/n_features. In the parameter setting of Logistic Regression, the solver was set as 'liblinear' and the penalty was set as '12', which satisfied the Gaussian distribution.

Additionally, since we used three computers in the experiment to simulate the cloud, the edge, and the terminal, our system demonstrates some other advantages. The path decision system computed by our algorithm at the edge-side is essentially similar in average elapsed time to that computed by GPS after direct upload to the cloud. Thus, our system relieves about 1 MFLOPS of computing power for the cloud per use without additional waiting for users.

Table 1 gives the results of these models. Although the decision tree is a relatively old algorithm, it maintains a higher accuracy rate in this field.

**Table 1.** Algorithm accuracy.

| Model | RMSE | AUR | F1 | VAR |
|---|---|---|---|---|
| **Decision Tree** | **0.1863** | **0.9573** | **0.9237** | **0.9653** |
| **SVM** | 0.2319 | 0.9165 | 0.9168 | 0.9462 |
| **Logistic Regression** | 0.2634 | 0.9377 | 0.9127 | 0.9306 |

Next, we decompiled the route into a complete route that conforms to the map. By reading the full data at that time and presenting them on the map, we can get a complete navigation route, as shown in Figure 9.

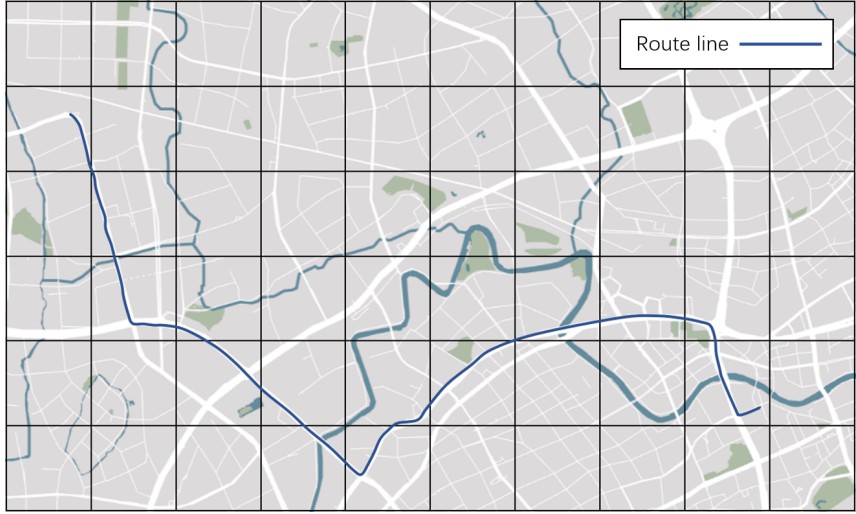

**Figure 9.** Final target route.

*4.6. Performance Analysis*

In addition to the cases shown in detail above, we conducted a total of 20 route decision experiments with different starting and ending points, premises, and requirements. We compared the results with the path decisions given by the real-time Google Maps for similar cases and found that our results are all consistent with those given by Google Maps. The Figure 10 shows the comparison between our results and Google Maps results for the above case.

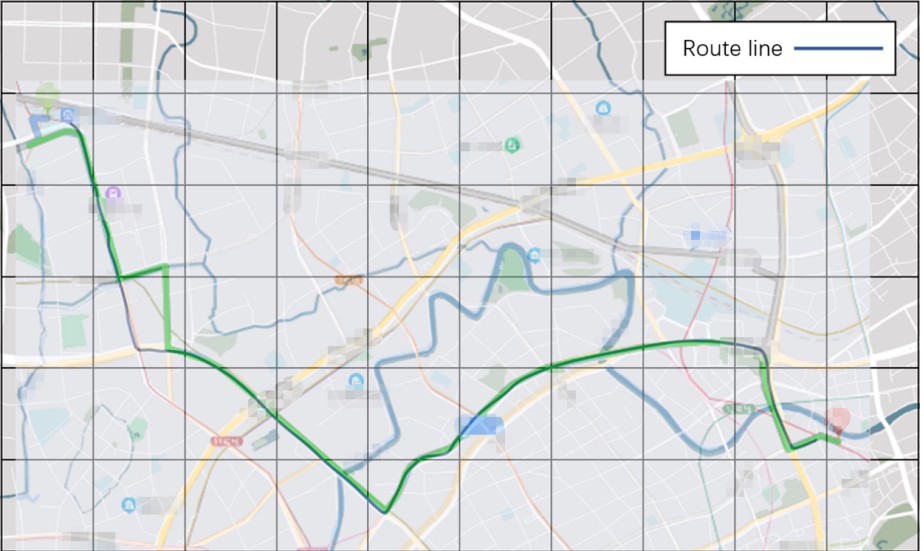

**Figure 10.** Comparison of the result.

In addition, we measured the time consumption of the system. We found that the computation time of the system is proportional to the distance between the start and end points. The longest distance of 20 km in our experiments took less than 30 s, which is acceptable in our opinion.

## 5. Conclusions

A new navigation algorithm with a methodology of multimodal fusion is proposed. Its core is a model based on the clustering-based decision tree, which makes statistics and analyzes historical data, considering the relationship and influence of each feature. Therefore, the model can provide a route selection algorithm with high accuracy and efficiency.

The model is modular, allowing the use of given conditions to seek needed results to meet the needs of users.

Historical data are aggregated data. When useful information is extracted from historical data, the information becomes knowledge. By designing and processing the data, the characteristics and information of different routes can be accurately estimated.

At the same time, the model is also used in the Spatio-temporal platform which is very popular in the field of transportation recently. Within the framework of edge computing, huge computing and tasks are offloaded to edge devices. Meanwhile, data are divided into different geographical areas and periods. Thus, the model is made within a specific range of data. Additionally, the speed of data processing is greatly accelerated.

Our model also has some unresolved problems, including determining the optimization structure of the model and the inability to improve the route of some real-time emergencies. In the future, it will be interesting to consider reinforcement learning with various external factors in traffic flow prediction and route navigation.

**Author Contributions:** Conceptualization, Y.L. and L.Z.; methodology, Y.L.; software, Y.L. and Q.X.; validation, Y.L. and Q.X.; formal analysis, Y.Z. and W.F.; investigation, Y.Z., W.F. and T.S.; resources, L.Z.; data curation, Y.L. and L.Z.; writing-original draft preparation, Y.L.; writing—review and editing, Y.L. and L.Z.; visualization, Y.L. and Q.X.; supervision, L.Z.; project administration, L.Z.; funding acquisition, L.Z. All authors have read and agreed to the published version of the manuscript.

**Funding:** This research was funded by Shanghai Collaborative Innovation Research Center for Mul-ti-network & Multi-modal Rail Transit grant number 20511106400.

**Informed Consent Statement:** Informed consent was obtained from all subjects involved in the study.

**Data Availability Statement:** The data presented in this study are available on request from the corresponding author. The data are not publicly available due to the data provider's request for confidentiality of the data and results.

**Acknowledgments:** The work was sponsored by Shanghai Collaborative Innovation Research Center for Multi-network and Multi-modal Rail Transit and partly by Shanghai Science and Technology Innovation Action Program No. 20511106400. We would like to thank Lei Zhang, DiDi Global Inc., and the National Transportation Information Co., Ltd. a lot for providing the data for us to complete this research.

**Conflicts of Interest:** The authors declare no conflict of interest.

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
