# Peer review of "Clustering-Based Decision Tree for Vehicle Routing Spatio-Temporal Selection"

_electronics, doi:10.3390/electronics11152379_

Round 1

Reviewer 1 Report

The article needs following improvements:

1) State specific research question/s at the end of introduction

2) There must be a section/sub-section on limitations

3) The abstract must be improved to include the details of the results

4) Section 3 must be renamed to Methods 

Author Response

Dear reviewer,

We would like to thank you a lot for your careful reading, helpful comments, and constructive suggestions, which have significantly improved the presentation of our manuscript.

We have carefully considered all comments from your review report and revised our manuscript accordingly. In the following section, we summarize our responses to each comment from the report.

Point 1:

- State specific research question/s at the end of introduction.

Response:

Thank you for pointing out this problem in our manuscript. We have added this part in the first section of our manuscript.

Point 2:

- There must be a section/sub-section on limitations.

Response:

Thank you for the above suggestion. We have added this part in the third section of our manuscript.

Point 3:

- The abstract must be improved to include the details of the results.

Response:

We thank you for pointing out this issue. We have rewritten the abstract part and have added the details of the results in the section.

Point 4:

- Section 3 must be renamed to Methods.

Response:

Thank you for the above suggestion. We have renamed the section 3.

We believe that our responses have well addressed all your concerns. And we hope our revised manuscript can be accepted for publication.

Yours sincerely,

Yixiao Liu

Reviewer 2 Report

The authors proposed a vehicle route spatio-temporal selection to improve vehicle routing performance and save cloud computing power. The proposed method looks reasonable, and the experimental results demonstrates the performance of the proposed method. However, there exist the following several problems in the paper, which reduces the paper quality.

1. The paper motivation in the introduction part is not strong. The authors list existing cloud real-time based route planning works and their disadvantages, which are good. However, the authors did not compare their proposed method and existing offline route planning works to show the advantages of the proposed method clearly. There exists several offline based route planning works and the authors may consider them as reference. The authors need to show it clearly.

[1] Diaz-Arango, G., et al. "Off-line route planner based on resistive grid method for vehicle guidance in real-time applications." 2018 IEEE 9th Latin American Symposium on Circuits & Systems (LASCAS). IEEE, 2018.

[2] Zhao, Yanli, Hao Zhou, and Yimin Liu. A Cost-Effective Offline Routing Optimization Approach to Employee Shuttle Services. No. 2017-01-0240. 2017.

2. Section 3 discussed the details about how to make route plans using the proposed route selection method. The authors only listed the basic steps in the selection process but there are limited contributions on the theoretical concept, which reduces the paper quality. The authors should spend enough space on explaining details in each step. Besides, the authors mentioned that they cluster the driving data with considering driving experiences. However, the authors did not explain which features are used as indexes of driving experiences and how the driving experiences are considered in the route selection process.  

3. In the experiment part, the authors made performance comparisons between their method and other methods. However, the other methods including SVM and logistic regression in the paper are too general and not representative, which makes their method less convincing. Besides, the authors only showed the experimental results based on a route example, it will be better if the authors provide evaluation results on large numbers of routes.

Author Response

Dear reviewer,

We would like to thank you a lot for your careful reading, helpful comments, and constructive suggestions, which have significantly improved the presentation of our manuscript.

We have carefully considered all comments from your review report and revised our manuscript accordingly. In the following section, we summarize our responses to each comment from the report.

Point 1:

- The paper motivation in the introduction part is not strong. The authors list existing cloud real-time based route planning works and their disadvantages, which are good. However, the authors did not compare their proposed method and existing offline route planning works to show the advantages of the proposed method clearly. There exists several offline based route planning works and the authors may consider them as reference. The authors need to show it clearly.

Response:

Thank you for pointing out this problem in our manuscript. We have rewritten the introduction part and added some comparisons and references with other offline route selection in the section of related works.

Point 2:

- Section 3 discussed the details about how to make route plans using the proposed route selection method. The authors only listed the basic steps in the selection process but there are limited contributions on the theoretical concept, which reduces the paper quality. The authors should spend enough space on explaining details in each step. Besides, the authors mentioned that they cluster the driving data with considering driving experiences. However, the authors did not explain which features are used as indexes of driving experiences and how the driving experiences are considered in the route selection process.

Response:

Thank you for the above suggestion. We have added some details about to this part in the third section of our manuscript.

Point 3:

- In the experiment part, the authors made performance comparisons between their method and other methods. However, the other methods including SVM and logistic regression in the paper are too general and not representative, which makes their method less convincing. Besides, the authors only showed the experimental results based on a route example, it will be better if the authors provide evaluation results on large numbers of routes.

Response:

We thank you for pointing out this issue. Since we did not have many algorithms to compare, we had to add many experiments to our own system and added the results in the fourth paragraph.

We believe that our responses have well addressed all your concerns. And we hope our revised manuscript can be accepted for publication.

Yours sincerely,

Yixiao Liu

Round 2

Reviewer 1 Report

I only have one suggestion. 4.6 should be renamed to something else. "More results of the system" is not an appropriate heading.

Author Response

Dear reviewer,

We would like to thank you a lot for your careful reading, helpful comments, and constructive suggestions, which have significantly improved the presentation of our manuscript. Your approval of our manuscripts is also greatly appreciated.

Point 1:

- I only have one suggestion. 4.6 should be renamed to something else. "More results of the system" is not an appropriate heading.

Response:

Thank you for pointing out this problem in our manuscript. We renamed the last paragraph of the last section and enriched the content of this paragraph.

Thank you again for your approval of our manuscript. We hope our revised manuscript can be accepted for publication.

Yours sincerely,

Yixiao Liu

Reviewer 2 Report

Thanks for the authors' responses and I have no other comments!

Author Response

Dear reviewer,

We would like to thank you a lot for your careful reading, helpful comments, and constructive suggestions, which have significantly improved the presentation of our manuscript. Your approval of our manuscripts is also greatly appreciated.

Due to another reviewer's comments on our manuscript, we renamed the last paragraph of the last section and enriched the content of this paragraph.

Thank you again for your approval of our manuscript. We hope our revised manuscript can be accepted for publication.

Yours sincerely,

Yixiao Liu